# Personal Cooling Garments: A Review

**DOI:** 10.3390/polym14245522

**Published:** 2022-12-16

**Authors:** Song Ren, Mengyao Han, Jian Fang

**Affiliations:** College of Textile and Clothing Engineering, Soochow University, Suzhou 215006, China

**Keywords:** personal cooling garments, thermal comfort, ice cooling, phase change material cooling, radiative cooling, air cooling, liquid cooling, thermoelectric cooling

## Abstract

Thermal comfort is of critical importance to people during hot weather or harsh working conditions to reduce heat stress. Therefore, personal cooling garments (PCGs) is a promising technology that provides a sustainable solution to provide direct thermal regulation on the human body, while at the same time, effectively reduces energy consumption on whole-building cooling. This paper summarizes the current status of PCGs, and depending on the requirement of electric power supply, we divide the PCGs into two categories with systematic instruction on the cooling materials, working principles, and state-of-the-art research progress. Additionally, the application fields of different cooling strategies are presented. Current problems hindering the improvement of PCGs, and further development recommendations are highlighted, in the hope of fostering and widening the prospect of PCGs.

## 1. Introduction

Thermal comfort is vital to human beings, not only because we are homothermic and need a consistent body temperature to survive, but also because thermal discomfort can lead to psychological problems, such as heat stress, which can result in tedium and exhaustion in mentality and physicality, causing personal health issues and the curtailment of work productivity [1,2]. According to a statement from the American Society of Heating, Refrigeration, and Air-Conditioning Engineers (ASHRAE), thermal comfort is a mind condition that conveys satisfaction with the ambient environment temperature [3]. Generally, the narrow range of comfortable temperature for human beings is between 20 °C to 27 °C [4]; outside of this range, we would need additional means, such as garments or air-conditioning, to maintain thermal comfort. 

Garments are a paramount part of personal thermoregulation, which can directly influence the heat-exchanging process between the human body and the environment [5], although the human body has a certain ability to automatically regulate its temperature through various activities, such as altering the rates of metabolism and blood flow, sweating, pore-shrinking, etc. [4] When the human body is in hot weather, the sweating and blood flow rates can be increased to enhance thermal diffusion. However, during extreme weather or harsh conditions, which exceed the ability to self-regulate, it is necessary to utilize functional garments to achieve thermal comfort. 

The personal thermal garment is an auspicious technology that can avoid thermally-induced health issues and provide a sustainable solution to reduce energy consumption [6]. With the global warming effect, there has been a rising demand for personal thermal comfort in high temperature weather. Energy consumption has been increasing continuously and dramatically, and is expected to triple by 2050 [7]. Therefore, it is urgent to meet the demand for the development of advanced thermoregulation garments. In consideration of these reasons, over the past few decades, a considerable amount of research of advanced thermal devices has been conducted to design and fabricate intelligent thermoregulation garments. At present, personal heating garments have enjoyed rapid development, from heating materials to heating techniques, strategies, and heating effect evaluation, and they have captured a dominant share of the thermal regulating textile market. There are various heating garments using different heating materials, such as carbon nanotube (CNT) [8,9], carbon fiber (CF) [10,11], graphene [12,13], metallic nanowire meshes [14,15], etc. Thermal regulating garments with cooling functions are also strongly demanded in numerous application conditions, including exercising or working in hot weather, personal portable cooling equipment in the medical area, or in special protective garments fields, such as astronaut garments, medical protective garments, and firefighter garments. However, compared with the large-scale commercial production of personal heating garments, the progress of personal cooling garments (PCGs) has been lagging behind. 

The concept of personal cooling garments was first put forward in 1958 [16]. However, it was not until 1962 that the first personal refrigeration garment was produced for the aerospace industry. Since then, many different kinds of cooling garments, such as air-cooled, liquid-cooled, radiative-cooled, and thermoelectric (TE)-cooled garments, have continuously been fabricated and tested. Due to the technical difficulty in achieving a highly efficient cooling effect on garments, the PCG market has developed at a much slower pace. 

In recent years, with the rapid evolution of smart and functional textiles, as well as the global demand of energy-saving solutions, more efforts have been devoted to the development of PCGs, with many new materials, cooling techniques, and device structure optimizations being examined. Therefore, a systematic review of recent advances in personal cooling garments is in timely demand, which needs to cover the classification and the forefront development of personal cooling materials and strategies in detail. In this review, we have focused on the state-of-the-art developments of personal cooling garments. We divide cooling garments into two categories based on whether electric power is required for the cooling functional: non-electric cooling and electric cooling. As shown in Figure 1, non-electric cooling contains ice cooling, phase change materials (PCMs) cooling, and radiative cooling, while electric cooling techniques include TE cooling, liquid cooling, and air-cooling. Furthermore, we summarized the major challenges of the development of PCGs and pointed out the future perspectives, trying to facilitate the progress of personal cooling garments.

## 2. Non-Electric Cooling 

A stable core body temperature is essential for maintaining optimal functions of the human body. To reduce the health risks caused by hazardous heat environments, microclimate cooling technologies have been developed to enhance the heat exchange between the human body and the environment. Non-electric cooling strategies, which do not require a power supply system [17,18,19], can be divided into three main types: ice cooling [20], PCMs cooling [21], and radiation cooling [22].

### 2.1. Ice Cooling 

In the early development of PCGs, an ice cooling garments [23] was one of the most common types due to the simplicity of preparation and low cost. It was designed with specific pockets to hold the ice, as shown in Figure 2. Through the ice melting process, the heat generated from the skin surface is absorbed, resulting in a temperature drop and thermal comfort in hot weather. Based on the high latent heat, availability, and low cost, the ice cooling garment was widely applied in protective garment industries [24,25], athletic wear [26,27,28], and military uniforms [29,30]. Juhani et al. confirmed that an ice vest effectively cooled the skin temperature, which was beneficial to both physiology and subjectivity [25]. Furthermore, the thermal comfort provided by the ice cooling vest remarkably increased work efficiency by 10%. The same conclusion was drawn by Cooter et al. [31]. From their study, the effectiveness of the ice vest in improving endurance performance during sustained heavy work was closely examined. However, contact with ice for a long time can lead to tissue irritation [32]. Additionally, the heavy weight, large volume, and cooling interruption have strongly restricted the application of ice cooling garments in daily life. Therefore, the investigation of flexible cooling sources represents a potential approach to solve these problems. Gel ice cooling technology [33,34,35] has been extensively explored lately as a promising candidate. Dehghan et al. compared the effectiveness of an ice gel cooling vest under two exercise intensities and demonstrated a noticeable skin temperature drop, and lower heat strain score index during light activities as a result of the cooling vest [33]. It is noteworthy that under heavy activities, there were no significant differences. Moreover, Chesterton et al. compared the skin-cooling effects of flexible frozen gel and frozen peas [34]. The result indicates that flexible frozen gel is not as effective as frozen peas in skin freezing and calming. Therefore, ice cooling garments with satisfactory cooling performance and flexibility is still a challenge, which requires further studies on advanced materials and technologies.

### 2.2. PCMs Cooling

PCMs can directly use the latent heat from the body or the environment to lower the temperature of the microclimate between the garment and the body, without any extra energy consumption [36]. With the exploration of materials, the inherent defects of the above-mentioned frozen materials (as a typical phase change material) have been revealed gradually, such as an uncontrollable cooling effect and limited cooling duration. Given these circumstances, the demand for seeking more comfort and effective PCM for wearable cooling application has been ceaselessly rising [37,38,39]. Li et al. developed a novel PCM cooling garment with a designed placement of PCM through multi-scenario experiments, which is light and convenient for daily use [40]. With the calculation method for the minimum charge of PCM, the final cooling garment is only 1.39 kg with obvious cooling performance, which exhibited a maximum 1.78 °C temperature drop during indoor walking. Besides packaging a PCM module in the garment, it is feasible for PCM to compound with yarns in various ways, such as impregnation and filling with hollow fibers [41], melt spinning [42], coating [43], and microcapsule encapsulation [44]. In the last decade, using these properties of PCMs to fabricate adaptive textiles or fibers has been conspicuous and noticeable [45,46,47]. V. Skurkyte-Papieviene et al. explored a type of PCM microcapsules MPCM32D, adding multiwall carbon nanotubes (MWCNTs) and poly (3,4-ethylenedioxyoxythiophene) poly (styrene sulphonate) (PEDOT: PSS) as thermally conductive additives to improve the outer shell thermal conductivity, which relatively enhanced the heat storage and release capability of PCMs microcapsules [44]. Yang et al. developed a 3D-printed flexible phase-change nonwoven fabric with excellent stability and durability [46]. Owing to the poly (ethylene glycol) (PEG)-grafted TPU prepolymer and embedded single-walled carbon nanotubes, the nonwoven fabric exhibited adequate thermal regulation and radiation resistance performance, even at cycles up to 2000. Moreover, a dramatic breakthrough in the commercial application of wearable PCM has been made by Outlast^®^ technologies [48]. The Outlast^®^ fiber exhibits the desired thermo-regulating performance through incorporating PCM by microencapsulation, remarkably reducing sweat formation by 48% in a hot temperature environment. In summary, the extraordinary breathability and flexibility ensure the PCM textile has a large-scale implementation in PCM, which is the critical development direction of PCM cooling garments [47]. 

### 2.3. Radiation Cooling 

Based on the highly transparent atmospheric window in the infrared (IR) wavelength range between 8 to13 μm, which coincides with the thermal radiation of humans, radiation cooling (RC) can dump human heat directly into outer space without external energy [49]. As a promising alternative technology to locally controlling skin temperature, RC has aroused wide interest in personal cooling garments by taking advantage of an atmospheric window [50,51,52]. Hsu et al. developed an RC textile with good transparency to mid-infrared human body radiation based on nanoporous polyethylene (nanoPE) substrate, which can be used as an infrared radiation heat dissipation fabric for an individual cooling strategy [51]. Due to nanoPE having high transparency and transmittance, it can be appropriately used as a heat dissipation material. Their textile promoted effective radiative cooling and lower temperature around 2.7 °C, which provides a promising textile for personal cooling garments. After development of their textile, they utilized nanoPE embedded with a bilayer emitter to fabricate a dual-mode textile, which has both cooling and heating modes just by flipping the textile, as shown in Figure 3 [53]. The bilayer emitter with the carbon side (high emissivity) and the copper side (low emissivity) noticeably provided a 6.5 °C temperature difference in artificial skin by flipping the same textile, showing a significant technological advancement in wearable RC application. 

However, when there are other intense thermal radiation sources, such as direct sunlight in hot weather, it is critical to investigate RC textiles with high reflection to the solar radiation band [22,54,55]. Because the solar spectrum is mainly between 0.3 to 4.0 μm [56], it is greatly prospective and feasible to both enhance solar reflection and dissipate human thermal radiation. Cai et al. proposed the first spectrally-selective RC textile for outdoor cooling with more than 90% reflection of solar irradiance and satisfactory transmission of the human body thermal radiation, which enabled simulated skin to avoid overheating by 5–13 °C compared with cotton fabric [57]. Irfan et al. developed a nanofabric by nanoparticle-doped polymer (zinc oxide and polyethylene) materials and electrospinning technology, which offered 91% solar reflectivity and 81% mid-infrared transmissivity, resulting in a 9 °C cooling performance compared with cotton textile [58]. Through numerous studies, it has been solidly confirmed that RC garments have a great potential for mass adoption of effective and energy-saving cooling technology in daily life. 

## 3. Electric Cooling 

With the vigorous developments of wearable electronic devices, the utilization of electronic devices with cooling materials and strategies is the main development direction of intelligent cooling garments. Under these circumstances, the personal cooling garment under electric power supplied has enjoyed a sustained evolution. At present, the main types of this area are air cooling garments (ACGs) [59], liquid cooling garments (LCGs) [60], and TE cooling garments [61].

### 3.1. Air Cooling 

As a traditional cooling strategy, an ACG provides thermal comfort by forcing air to flow through the microclimate between the clothing and the human body [62]. The major advantages of ACGs are low cost, light weight, and portability [63]. At the early stage of development, ACGs were applied in aerospace and military fields, such as protective garments for pilots and soldiers to reduce heat stress [64,65,66]. Hadid et al. reported that during the same intensity of activity, an ACG led to a lower body temperature and reduction of perspiration by 20% [67]. However, more complex living and working conditions require higher cooling performance with better comfort and security of the ACG. Additionally, ACG configurations are normally heavy and large, limiting their widespread applications in our daily life. Therefore, optimization of wearing comfort and thermal comfort is in great demand [68,69]. Yang et al. investigated the influence of clothing size and the air ventilation rate on the cooling performance of ACGs, which demonstrated that air ventilation greatly reduced the predicted core temperatures in two garment sizes; however, there was almost no impact of garment size on the predicted thermophysiological responses in high ventilation [70]. Similarly, Zhao et al. enhanced the cooling performance through clothing eyelet designs, which provided an alternative method to optimize ACGs in practical use [71]. 

The cooling effect provided by a single air-cooling strategy can be further improved to satisfy the ever-growing cooling needs. Consequently, a lot of efforts have been made in studying the possibility of combining an air-cooling strategy with other cooling methods to explore novel cooling approaches. Ni et al. developed a novel hybrid personal cooling vest (PCV), as shown in Figure 4a [72]. Their novel PCV was incorporated with PCMs and ventilation fans, which indicates the applicability and reliability of this hybrid cooling garment. Through experimental studies, the cooling efficacy in a hot, humid climate chamber was examined. Lou et al. investigated the relationship between the cooling effect with different body positions based on an air tubing network and TE cooling plates, which is helpful to improve the combination of the cooling system and the garment in an effective and comfortable way in daily life [73]. Based on large numbers of studies, the ACG is verifiably suitable to meet the practical wearing demand based on the notable portability and simplicity of operation and usage. 

### 3.2. Liquid Cooling 

Liquid cooling garments are generally embedded with circulating water tubes filled with a cold liquid resource and a micro water pump device at the inner layer to drive the liquid flowing in the tube to reduce the temperature [65,74], as shown in Figure 5a. The first LCG was supplied to an astronaut as a protective garment to lower the body temperature [75]. After numerous explorations, the LCG has been proven to be one of the most promising technologies in the wearable cooling arena and is used in many fields, such as military [76], mining [77], and sporting [78]. Guo et al. proposed a heat transfer model of LCGs to analyze the effects of different factors on the LCG performance and optimize the design of LCGs [74]. With the optimization they made, the max cooling rate reached 243.2 W/m^2^ with a maximum work duration time of 3.36 h. However, due to the embedded heavy device and the risk of skin burn caused by the stream formation [79], the LCG needs more efforts to modify the cooling systems for safety and comfort. Grazyna et al. developed a novel LCG with a sensor to adjust the microclimate temperature, and modular knitted fabric, which can be directly worn on the human body [80], resulting in great safety and comfort for the user. The notable tube system they proposed fitted to the skin nicely, which enhanced the thermal conductivity and cooling efficiency. Shu et al. proposed an intelligent temperature control system in LCGs [81], which verified that the new smart system can regulate temperature accurately and extend the duration of more than 30% of the cooling devices. 

To achieve significant amelioration of the LCG, combining other cooling technologies with LCGs exhibits remarkable potential in practical usage [55]. Zhang et al. reported a novel LCG with TE materials, which considerably alleviates the thermal stress hazard, as shown in Figure 5b [82]. They explored the effect of ambient temperature and heat dissipation on cooling performance, which also proposed an accurate method of assessing LCGs. 

### 3.3. TE Cooling

For individual wearable cooling devices, air cooling and liquid cooling garments are inconvenient and heavy with bulky air or fluidic channels, and the cooling performance of them is not stable and reliable [83]. Given these intrinsic disadvantages, homeostatic solid-state cooling strategies, such as electrocaloric cooling [84,85], magnetocaloric [86], and TE cooling [87], have garnered significant attention. Particularly, TE cooling exhibits great potential for practical industrial cooling application owing to its reliable cooling performance and small device dimensions [88]. Therefore, wearable TE cooling provides a desired alternative method for personal thermal comfort [89].

TE materials consisting of different types (N- and P-type) of conductors or semiconductors can be used to directly convert heat energy to electricity and vice versa [90]. Based on the Peltier effect [91,92], when a direct current passes through different TE materials, heat is absorbed or dissipated at the junctions, resulting in a hot side and a cold side [93]. TE devices are lightweight and have no moving parts or noise, which can be considered as the cooling strategy with the most potential in the future [94].

For small solid-state cooling, the TE cooling plate (TECP) is a good solution for industrial TE application due to their large-scale commercial production, low cost, and light weight [95]. Therefore, cooling devices based on TECP were conceived to explore practical applications of TEC, such as the Embr wave bracelet and Sony Reno pocket that can be worn on the arm or put in the pocket of a garment to cool the connected skin. Luo et al. successfully embedded a TECP cooling system in an undergarment, which has dramatic light weight and portability [96]. As shown in Figure 6, the TE cooling module embedded with heatsink as a cooling source, connected with a tubing network to provide uniform and sufficient cooling performance, resulting in 15% energy savings of indoor heating, ventilation, and air conditioning (HAVC). However, the rigid and bulky heatsink greatly reduces the wearability of TE and obstructs the development of flexible TE cooling technology. Therefore, a flexible heatsink has been predicated as a hopeful approach to ameliorate the flexibility of TE cooling. Jaeyoo et al. proposed a flexible cooling device with a designed heatsink that was composed of silicone elastomer, phase change material, and graphite powders. Their device could lower temperatures by around 5 °C and maintained temperatures for more than 5 h under ambient air temperature with commercial TECPs. Recently, an innovative mask integrated with thermoelectric devices and a 3D printed framework has been reported [97]. Through the test, a notable reduction in the temperature was around 3.5 °C with a low voltage application. 

Nevertheless, more efforts need to be made to seek efficient and wearable thermoelectric devices with prospective flexibility. Therefore, fabricating and softening the TEC structure with a flexible heatsink is clamant to be explored. Hong et al. developed a flexible and portable thermoelectric cooler that has a scalable application, as seen in Figure 7 [98]. Inorganic semiconductor thermoelectric pillars were placed on two stretchable flexible Ecoflex films, which were filled with thermally conductive filler aluminum nitride to improve thermal conductivity. The device could work continuously for 8 h without any heatsink equipment and delivers more than 7.6 °C cooling effects. Zhang et al. proposed a wearable TEC based on a two-layer flexible heatsink [99], which was composed of hydrogel and nickel foam as phase change material to absorb the heat, and thermally conductive material to conduct heat, respectively. Furthermore, the discrete heatsink they explored ensured the desired flexibility and achieved a large temperature drop of 10 °C under 0.3 A input current.

To further ameliorate flexibility and promote the flexible TEC evolution, TE fiber has been widely investigated to provide better wearing comfort. Zhang et al. prepared super long flexible TE micro/nanowires by hot drawing technology [100]. They fabricated both N- and P-type semiconducting materials into a fiber and covered them with borosilicate glass to protect and prevent the fiber from being oxidized. The fiber was flexible and had a high TE property, and could be easily woven into the textile and create 6.2 °C cooling performance. Zheng et al. reported a novel design of fiber-based thermoelectric textiles (TETs) [101]. Their TE fiber was fabricated by inorganic TE materials and liquid metal, which were encapsulated by polydimethylsiloxane. The TET provided a stable cooling performance of 3.1 °C with notable stretchability and flexibility. 

## 4. Conclusions

This work has summarized the latest development of PCGs and reviewed the existing research on cooling strategies and materials. Depending on the requirement of electric power, PCGs can be divided into non-electric and electric cooling garments. Each type of PCGs has its intrinsic advantages and drawbacks. As for non-electric cooling garments, the primary advantage is that they are energy-saving technologies, but the cooling performance is generally lower than electric cooling. However, with a power source, the portability tends to decrease, and there is a more complex requirement on system integration. The detailed comparison of different PCGs is shown in Table 1.

The use of PCGs can significantly improve human comfort and health during various warm conditions. It also contributes to global efforts in saving energy and reducing environmental pollution; however, with the increasing complexity of environmental change and people’s demands, more and more factors need to be considered when designing and evaluating cooling garments. These factors need to be comprehensively explored from the three dimensions of environment, clothing, and the human body. In terms of environment, different use scenarios should be considered, such as changing ambient temperature and humidity. From the clothing itself, the selection of fabrics, the weight, size, location, and structure of the cooling device, the combination of fabrics, and the overall design of clothing need to be studied more. For the human, the thermal balance ability of the human body itself, the different cold and heat perception abilities of different parts of the human body, and even different postures will affect the cooling effect of the refrigerated clothing. Therefore, numerous variables affect the design and performance of personal cooling garments. 

Nevertheless, there are still restrictions existing in the developments so far. Thus, firstly, it is paramount to improve the cooling performance of the materials and explore the new advanced strategies, which are supposed to be more light, low-cost, environment-friendly, and easy to prepare. Furthermore, optimizing and modifying the personal thermal comfort evaluation system is required to consider more parameters to assess the cooling efficiency of different PCGs. Simultaneously, broadening the application fields is another crucial element requiring further development, so that these garments can not only be used in special protective garment fields, such as aerospace, but also in personal daily life, including refrigeration sportswear, personal portable medical cooling devices, etc. Apart from this, some of the PCGs only exist in laboratories, and large-scale commercial production should be developed in the near future. Finally, studying and analyzing the novel structures of PCGs with the consideration of enhanced wearability may be strongly needed for future development of personal cooling garments. 

## Figures and Tables

**Figure 1 polymers-14-05522-f001:**
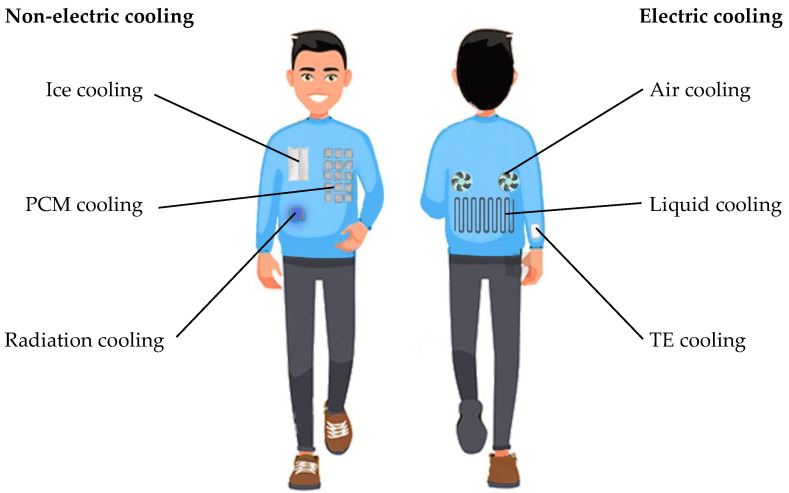
Cooling strategies of personal cooling garments.

**Figure 2 polymers-14-05522-f002:**
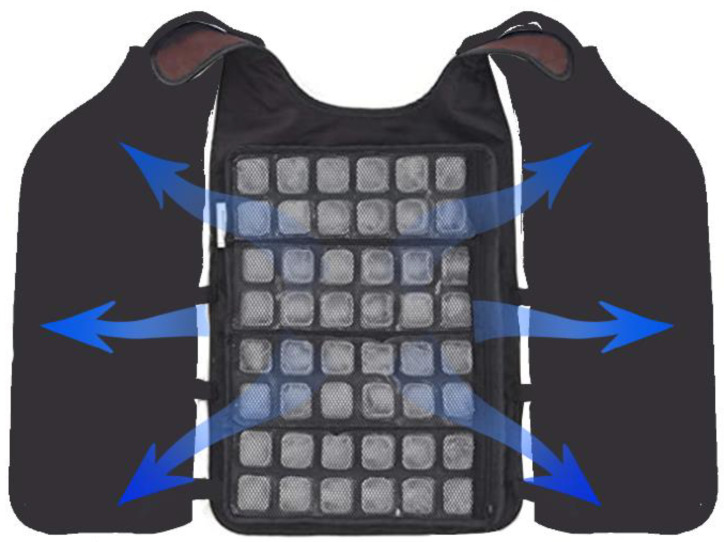
Illustration of an ice cooling vest.

**Figure 3 polymers-14-05522-f003:**
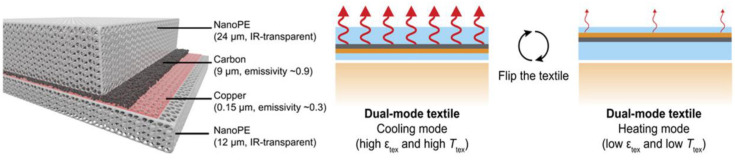
Layered structure of the textile and schematic diagram of the two modes, reproduced from [53].

**Figure 4 polymers-14-05522-f004:**
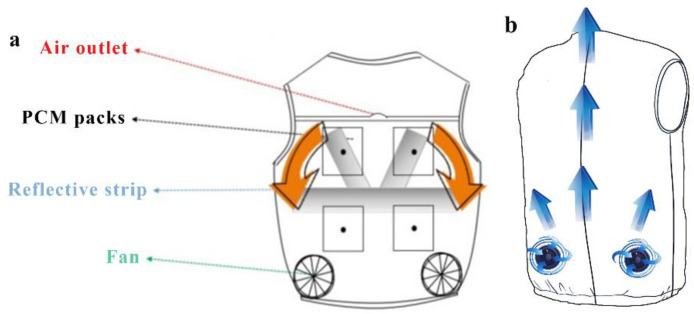
(**a**) Illustration of air cooling with a PCM cooling vest (PCV), adapted from [72], (**b**) Schematic diagrams of an air cooling garment.

**Figure 5 polymers-14-05522-f005:**
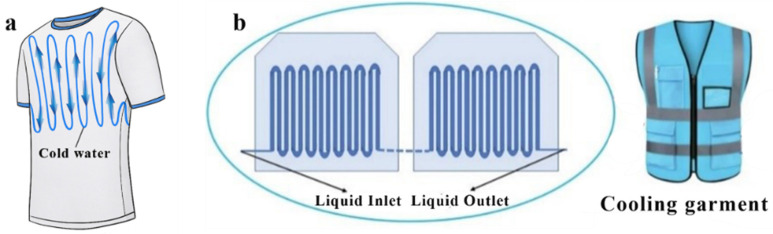
(**a**) Schematic diagrams of a liquid cooling garment. (**b**) Liquid cooling garment with a TE device, reproduced or adapted from [82], with permission from Elsevier, 2022.

**Figure 6 polymers-14-05522-f006:**
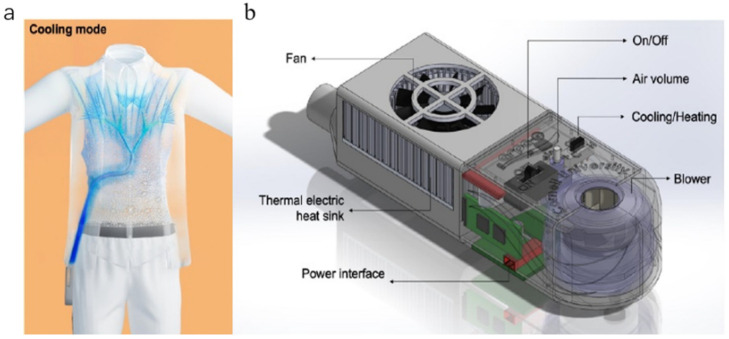
Schematic image of (**a**) a TE cooling undergarment and (**b**) illustration of a TE cooling module, reproduced or adapted from [96], with permission from Elsevier, 2020.

**Figure 7 polymers-14-05522-f007:**
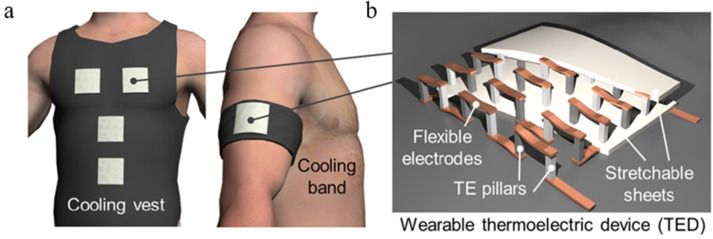
Schematic images of (**a**) cooling garments with wearable TE devices (TEDs) and (**b**) the structure of a TED, [98].

**Table 1 polymers-14-05522-t001:** The summary of different cooling strategies.

Cooling Strategy	Mechanism	Advantages	Disadvantage(s)
Ice cooling	Ice absorbs the body heat during melting to cool it down.	High cooling performance; no consumption of energy; reach −20 °C low temperature [19]	Limited cooling duration; heavy and no flexibility; penetration of condensate; risk of tissue damage
PCMs cooling	PCMs absorb heat through a phase change process when the temperature rises to a certain range.	No consumption of energy; easy maintenance; cooling efficiency: 1.78–3 °C temperature drop of skin [40,102]	High cost of materials and manufacturing; limited cooling duration
Radiative cooling	High solar reflection and infrared emissivity to prevent heat absorption and enhance self-heat release	No consumption of energy; light and breathable; cooling efficiency: temperatures 2–13 °C lower than normal textile [51,57]	High cost of materials and manufacturing
Air cooling	Blowing air into a garment to enhance sweat evaporation	Long-term cooling; easy manufacturing; cooling efficiency: 0.3–1 °C temperature drop of skin [59,103]	The impermeable fabric aggravates thermal discomfort
Liquid cooling	Water tubes in the garment circulate cooled water and lower the temperature	Long-term cooling; easy manufacturing; cooling efficiency: 1.2–2.5 °C temperature drop of skin [104]	Large and heavy devices; poor comfort; risk of security
TE cooling	When direct current passes through circuits composed of different semiconductors, heat is absorbed or dissipated at the junctions, resulting in a hot side and a cold side	Stable, reliable, and regulated cooling performance; small and light; cooling efficiency: 38 °C temperature drop of skin [98,101]	High cost of materials and manufacturing; no commercial flexible devices

## Data Availability

The study did not report any data.

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
