# Peer review of "Personal Cooling Garments: A Review"

_polymers, 2022, doi:10.3390/polym14245522_

Round 1
Reviewer 1 Report
This review provided a solid and profound summary of two significant Personal Cooling Garments (PCGs) categories with details on their cooling mechanisms, up-to-date development status, and sufficient references. The PCGs field needs a fresh and integrated review paper for researchers who are new and interested in this area to understand the overall development of PCGs. I consider this review a very good introductory material for beginners, as some highly cited PCGs review papers were published around ten years ago.
However, there are some issues I would like to raise:
1. It concisely described the mechanisms of different PCGs with figures. However, only four reference papers (from all five figures) were used to describe at least six cooling strategies. Schematic diagrams were used in Figure 4&5, preferably similar images should be used in Figure 2&3. The reader may benefit from a visualization way of demonstration.
2. The comparison among active/passive cooling strategies could be expanded as only a summary table was given. Some descriptions look identical, e.g. 'High cost of materials and manufacturing; limited cooling efficiency' appeared twice in the summary table, which remained unclear how exactly the drawbacks of different cooling strategies are different. The author may add details of comparison in the context or include some physical parameters (e.g. cooling rates) to quantitatively compare both advantages and drawbacks.
3. Some other possible directions to improve the background knowledge could be: giving a brief look at human thermoregulation; briefly introducing garment testing procedure, and a discussion on variables affecting the performance and efficiency of garments.
Author Response
Point 1: It concisely described the mechanisms of different PCGs with figures. However, only four reference papers (from all five figures) were used to describe at least six cooling strategies. Schematic diagrams were used in Figure 4&5, preferably similar images should be used in Figure 2&3. The reader may benefit from a visualization way of demonstration.
Response 1: Thank you very much for your suggestions. We added more papers and figures to descripe cooling strategies. Furthermore, figures with brief schematic diagrams have been modified in the revised manuscript accordingly.
Figure 2. Illustration of ice cooling vest
Figure 3. Layered structure of the textile and schematic diagram of two mode [53]
Figure 4. a. Illustration of the air cooling with PCM cooling vest (PCV) [72]. b. Schematic diagrams of air cooling garment.
Figure 5. a. Schematic diagrams of liquid cooling garment. b. Liquid cooling garment with TE device [82].
Point 2. The comparison among active/passive cooling strategies could be expanded as only a summary table was given. Some descriptions look identical, e.g. 'High cost of materials and manufacturing; limited cooling efficiency' appeared twice in the summary table, which remained unclear how exactly the drawbacks of different cooling strategies are different. The author may add details of comparison in the context or include some physical parameters (e.g. cooling rates) to quantitatively compare both advantages and drawbacks.
Response 2: Thank you very much for your comment. The explanation has been made in the table. We modified the description of cooling performance and limitations.
|
Cooling strategy |
Mechanism |
Advantages |
Disadvantage |
|
Ice cooling |
Ice absorbs the body heat during melting to cool down |
High cooling performance; no consumption of energy; reach -20 °C low temperature [19] |
Limited cooling duration; heavy and no flexibility; penetration of condensate; risk of tissue damage |
|
PCMs cooling |
PCMs absorb heat through phase change process when temperature rises to a certain range |
No consumption of energy; easy maintenance; cooling efficiency: 1.78-3 °C temperature drop of skin [40,102] |
High cost of materials and manufacturing; limited cooling duration |
|
Radiative cooling |
High solar reflection and infrared emissivity to prevent heat absorption and enhance self-heat release |
No consumption of energy; light and breathable; cooling efficiency: temperatures 2–13 °C lower than normal textile [51,57] |
High cost of materials and manufacturing |
|
Air cooling |
Through blowing air into a garment to enhance the sweat evaporating |
Long term cooling; easy manufacturing; cooling efficiency: 0.3-1 °C temperature drop of skin [59,103] |
The impermeable fabric aggravates thermal uncomfort; |
|
Liquid cooling |
Add water tubes into garment to circulate cooled water and lower the temperature |
Long term cooling; easy manufacturing; cooling efficiency: 1.2-2.5 °C temperature drop of skin [104] |
Large and heavy devices; poor comfort; risk of security |
|
TE cooling |
When direct current passes through circuit composed of different semiconductors, heat is absorbed or dissipated at the junctions, resulting in hot side and cold side |
Stable, reliable, and regulated cooling performance; small and light; cooling efficiency: 38 °C temperature drop of skin [98,101] |
High cost of materials and manufacturing; no commercial flexible devices |
Point 3. Some other possible directions to improve the background knowledge could be: giving a brief look at human thermoregulation; briefly introducing garment testing procedure, and a discussion n on variables affecting the performance and efficiency of garments.
Response 3: Thank you very much for your comment. We added a brief description of human thermoregulation at the first introduction part.
“Garment is a paramount part of personal thermoregulation, which can directly influence the heat-exchanging process between human body and the environment [5], although human body has a certain ability to automatically regulate the temperature through various activities, such as altering the rate of metabolic and blood flow, sweating, and pore-shrinking, etc. [4] When the human body was in hot weather, the sweating and blood flow rate can be increased to enhance thermal diffusion.”
Furthermore, the last paragraph of the summary part has been simplified into two sections for adding a brief look at background knowledge.“
“The use of PCGs can significantly improve human comfort and health during various warm conditions, it also contributes to global efforts in saving energy and reducing environmental pollution, however, with the increasing complexity of the using environment and people's demands, more and more factors need to be considered when designing and evaluating cooling garments, which needs to be comprehensively explored from the three dimensions of environment, clothing, and the human body. In terms of environment, different use scenarios should be considered, such as changing ambient temperature and humidity. From the clothing itself, the selection of fabrics, the weight, size, location, and structure of the cooling device, the combination with fabrics, and the overall design of clothing need to be studied more. For the human, the thermal balance ability of the human body itself, the different cold and heat perception abilities of different parts of the human body, and even the different postures will affect the cooling effect of the refrigerated clothing. Therefore, numerous variables affect the design and performance of personal cooling garments.
Nevertheless, there are still restrictions existing in the developments so far…”

Reviewer 2 Report
Page 3 line 128 must be V. Skurkyte-Papieviene and et al.
Page 4 line 150 must be human body radiation based on nanoporous polyethylene (nanoPE) substrate which can be
Page 8 line 293 must be both N and P type semiconducting materials into
1. What is the main question addressed by the research?
Authors summarized the current status of personal cooling garments (PCGs), depending on the requirement on electric power supply, by dividing into two categories with systematic instruction on the cooling materials working principles, and state-of-the-art research progress. The application fields of different cooling strategies were analyzes as well.
2. Do you consider the topic original or relevant in the field? Does it address a specific gap in the field?
The topic is relevant in the field.
3. What does it add to the subject area compared with other published material?
It provides a concentrated knowledge in the research field of PCGs that has already been done.
4. What specific improvements should the authors consider regarding the methodology? What further controls should be considered?
No specific improvements are required
5. Are the conclusions consistent with the evidence and arguments presented and do they address the main question posed?
Yes, the conclusions are correct.
6. Are the references appropriate?
Yes, it appropriate.
7. Please include any additional comments on the tables and figures.
I have no additional comments.
Author Response
Point 1: Page 3 line 128 must be V. Skurkyte-Papieviene and et al.
Page 4 line 150 must be human body radiation based on nanoporous polyethylene (nanoPE) substrate which can be
Page 8 line 293 must be both N and P type semiconducting materials into
Response 1: Thank you so much for your comment. All the detailed modifications have been made exactly.
1) V. Skurkyte-Papieviene et al. explored a type…(Page 4 line 139)
2) …human body radiation based on nanoporous polyethylene (nanoPE)…(Page 4 line 161)
3) …both N and P type semiconducting materials into…(Page 8 line 336)

Reviewer 3 Report
The manuscript is well written, and a few phrases in English can be improved as highlighted in the attached document.

Author Response
Point 1: The manuscript is well written, and a few phrases in English can be improved as highlighted in the attached document
Response 1: Thank you so much for your time and comments on our manuscript, we will do our best to improve the quality of the manuscript according to your comments and suggestions.
1) …is still a challenge, which requests further studies on advanced materials and technologies. (Page 3 line 115)
2)…to fabricate dual-mode textile, which have both cooling and heating mode just by flipping the textile. (Page 5 line 167)
3) …the personal cooling garment under electric power supplied has enjoyed a sustained evolution. (Page 5 line 193)
4) …has been wildly investigated…(Page 8 line 334)
5)……both N and P type semiconducting materials into…(Page 8 line 336)
